# Familial Risk and Heritability of Hematologic Malignancies in the Nordic Twin Study of Cancer

**DOI:** 10.3390/cancers13123023

**Published:** 2021-06-16

**Authors:** Signe B. Clemmensen, Jennifer R. Harris, Jonas Mengel-From, Wagner H. Bonat, Henrik Frederiksen, Jaakko Kaprio, Jacob v. B. Hjelmborg

**Affiliations:** 1Department of Epidemiology, Biostatistics, and Biodemography, Institute of Public Health, University of Southern Denmark, 5000 Odense C, Denmark; jmengel-from@health.sdu.dk (J.M.-F.); jhjelmborg@health.sdu.dk (J.v.B.H.); 2Danish Twin Registry, Institute of Public Health, University of Southern Denmark, 5000 Odense C, Denmark; 3Division of Health Data and Digitalisation, Norwegian Institute of Public Health, 0213 Oslo, Norway; jenniferruth.harris@fhi.no; 4Department of Clinical Genetics, Odense University Hospital, 5000 Odense C, Denmark; 5Department of Statistics, Paraná Federal University, Curitiba 81531-980, Brazil; wbonat@ufpr.br; 6Department of Haematology, Odense University Hospital, 5000 Odense C, Denmark; Henrik.Frederiksen@rsyd.dk; 7Department of Clinical Research, Institute of Public Health, University of Southern Denmark, 5000 Odense C, Denmark; 8Department of Public Health and Institute for Molecular Medicine Finland, University of Helsinki, 00014 Helsinki, Finland; jaakko.kaprio@helsinki.fi

**Keywords:** twin study, cumulative risk, familial risk, risk between different cancers, heritability, biometric modelling, hematologic malignancy

## Abstract

**Simple Summary:**

Hematologic malignancies account for 8–9% of all incident cancers. Both genetic and environmental risk factors contribute to cancer development, but it is unclear if there is shared heritability between hematologic malignancies. This study aimed to investigate familial predisposition to hematologic malignancies using the largest twin study of cancer in the world. We compared individual risk in the general population and the risk of cancer in one twin before some age given that the other twin had (another) cancer before that age. Furthermore, by analyzing information about whether the twins were identical or fraternal, we could estimate the relative importance of genetic and environmental influences on the risk for developing hematologic cancers. This study confirmed previous findings of familial predisposition to hematologic malignancies and provides novel evidence that familial predisposition decreases with increasing age. The latter points to the importance of taking age into account in the surveillance of hematological cancers.

**Abstract:**

We aimed to explore the genetic and environmental contributions to variation in the risk of hematologic malignancies and characterize familial dependence within and across hematologic malignancies. The study base included 316,397 individual twins from the Nordic Twin Study of Cancer with a median of 41 years of follow-up: 88,618 (28%) of the twins were monozygotic, and 3459 hematologic malignancies were reported. We estimated the cumulative incidence by age, familial risk, and genetic and environmental variance components of hematologic malignancies accounting for competing risk of death. The lifetime risk of any hematologic malignancy was 2.5% (95% CI 2.4–2.6%), as in the background population. This risk was elevated to 4.5% (95% CI 3.1–6.5%) conditional on hematologic malignancy in a dizygotic co-twin and was even greater at 7.6% (95% CI 4.8–11.8%) if a monozygotic co-twin had a hematologic malignancy. Heritability of the liability to develop any hematologic malignancy was 24% (95% CI 14–33%). This estimate decreased across age, from approximately 55% at age 40 to about 20–25% after age 55, when it seems to stabilize. In this largest ever studied twin cohort with the longest follow-up, we found evidence for familial risk of hematologic malignancies. The discovery of decreasing familial predisposition with increasing age underscores the importance of cancer surveillance in families with hematological malignancies.

## 1. Introduction

Hematologic malignancies account for 8–9% of all incident cancers, and the incidence has generally increased since the 1960s [1,2]. The major hematologic malignancies are Hodgkin lymphoma (HL), non-Hodgkin lymphoma (NHL), leukemia, and multiple myeloma (MM). The epidemiology and risk factors of these cancers have been studied for decades, but their etiology is still not entirely clarified. Heterogeneity within hematologic malignancy subtypes seems to pose a challenge for epidemiological identification of risk factors. Additionally, the incidence rates vary by age, sex, and geography [3,4,5,6,7,8,9].

A recent study of twins in the Nordic countries revealed that twin concordance for cancer often manifests across, rather than within, cancer types [10,11]. Namely, if one twin develops cancer, the co-twin has an increased risk of developing cancer but not necessarily at the same site. Previous epidemiological studies of hematologic malignancies suggest that this is indeed the case; they report evidence of familial predisposition across both hematologic and non-hematologic malignancies [12,13,14,15,16,17,18,19,20,21]. For instance, among women from Connecticut, lung cancer in first-degree relatives as well as breast and ovary cancer in siblings are associated with a significantly increased risk of NHL [12]. Moreover, the results indicate a stronger association between siblings than parents [9,12,13,14,15]. The impact of sex on familial predisposition has also been assessed, though the results differ by type of malignancy [9,14,15,16]. While the body of research indicates that familial effects for hematologic malignancies influence the risks, they are not able to further identify the source of these effects into those explained by shared genes and environment.

Twin studies allow for delving into the nature of this familial predisposition by partitioning the variation in risk into components explained by genetic and environmental influences. In 1996, Mack et al. [22] published findings from an American twin study suggesting genetic susceptibility to HL in young adulthood. A few years later, Lichtenstein et al. [23] conducted a study of twins from Denmark, Finland, and Sweden and reported familial influences for leukemia but did not take timing of events into account. A more recent Nordic study by Mucci et al. [10] also included Norwegian twins and had a longer follow-up time; they categorized leukemia as “acute” or “other” and provided an uncertain estimate of heritability for other leukemia: 57% (95% CI: 0–100%). Both of these Nordic studies analyzed multiple cancer types, including hematologic malignancies, but neither were able to estimate genetic and environmental variance for the remaining hematologic malignancies as no or too few concordant pairs were observed.

In this study, we utilize the world’s most comprehensive database of twins for studying cancer. We aim to explore the genetic and environmental contributions to variation in risk of hematologic malignancies and to characterize familial dependence within and across hematologic malignancies. We take into account timing of events, censoring, and competing risk of death.

## 2. Materials and Methods

The Nordic Twin Study of Cancer (NorTwinCan) is an international, multidisciplinary collaboration of researchers working to investigate the genetic and environmental underpinnings of cancer. The NorTwinCan cohort is based on nationwide twin registers linked to national cancer and mortality registers in Denmark, Finland, Norway, and Sweden [10,11,23]. Details about the twin registers in each of the four participating countries were described previously by Skytthe et al. [24]. Following the latest linkage update in 2018, the study now includes around 400,000 individual twins from four countries, out of which around 315,000 have known zygosity [11]. The analyses presented here are based on these 315,000 monozygotic (MZ) and dizygotic (DZ) twins. The DZ twin pairs can be of the same or opposite sex unless otherwise specified.

Cancer diagnoses were based on the International Statistical Classification of Diseases and Related Health Problems, Tenth Revision (ICD-10) and were grouped according to the NORDCAN classification system [25]. This study includes the main hematologic malignancies, i.e., Hodgkin lymphoma (HL) (C81), non-Hodgkin lymphoma (NHL) (C82–86), multiple myeloma (MM) (C90), and leukemia (C91–95). Due to inconsistencies among the classification of leukemia subtypes between Nordic countries, a sub-classification of leukemia was not carried out. In the Danish data, ICD-10 was used along with the International Classification of Diseases for Oncology, Third Edition (ICD-O-3) when identifying lymphoma incident cases, which might cause the cumulative incidences of HL and NHL to diverge from those of other Nordic countries. The incidence of cancer among twins has been demonstrated to mirror that of the general population [10,24].

### 2.1. Cumulative Incidence and Measures of Cross-Cancer Familial Dependence

For hematologic malignancies overall and separately, we estimated the lifetime risk, defined as risk of cancer diagnosis before age 100. The lifetime risks were obtained from the cumulative incidence function by age estimated using nonparametric counting process modelling, as described by Scheike et al. [26]. The lifetime risks were stratified by sex, zygosity, and cohort for representativeness comparison. Delayed entry determined by the date of register initiation and right-censoring caused by end of follow-up or emigration were accounted for by this approach. Likewise, competing risk of death was accounted for.

The nonparametric risk scale approach, described by Scheike et al. [27], allows for the estimation of cross-cancer familial dependence in terms of case-wise concordance and relative recurrence risk. The case-wise concordance is the conditional risk of cancer in a twin before some age given that the other twin developed (another) cancer before that age and was estimated using nonparametric counting process modelling, as described by Scheike et al. [27]. This is referred to as familial risk. Familial risks for DZ twins also apply to full siblings, as they are as genetically alike. Familial risk by age was estimated for overall hematologic malignancies, and lifetime familial risks were estimated for hematologic malignancies, overall and separately. The relative recurrence risk of cross-cancers was defined as the concordance of cross-cancers, that is, the risk of both twins in a pair having cancer (different cancers) before some age, relative to the risk of the two cancers occurring independently. The latter is given by the product of cumulative incidences. Cross-cancer relative recurrence risks were estimated within the hematologic malignancies and for overall hematologic malignancy vs. other cancer sites. Familial predisposition to cross-cancer is indicated by an increased familial risk among DZ twins compared to the cumulative incidence or a relative recurrence risk for DZ twins greater than one. Furthermore, the influence of shared genetic factors is indicated by a higher familial dependence among MZ twins compared to DZ twins [26]. Inference of the measures assume approximate normality in distribution of estimators and test of differences was performed using a Pepe–Mori type test derived in a way similar to that described by Scheike et al. [27].

### 2.2. Biometric Modelling

The overall magnitude of the genetic and environmental contributions to the variation in risk for each hematologic cancer, taking the timing of events into account, can be assessed by modelling the difference in MZ and DZ pair covariances to the total variance in cancer risk assuming equal margins in MZ and DZ twins. This was achieved parametrically using the bivariate biprobit model, that is, the classic liability-threshold approach, taking censoring into account by inverse probability weighting [28]. Tetrachoric correlations for MZ and DZ pairs and further variance components of the polygenic biometric model from quantitative genetics, the “ADCE model” [28,29], were obtained from this model. The components that contribute to variance are additive genetic (A), dominant genetic (D), shared environmental (C), and unique environmental (E) effects. Heritability is then defined as the proportion of variation in liability to develop cancer attributed to genetic factors, that is, the sum of A and D. Shared environmental effects are those experiences and exposures that affect both twins in a pair. Unique environmental effects influence one twin but not the other, but it also includes measurement error as well as random effects and genetic differences even between MZ pairs. Hence, environmental effects in this setting refer to external non-genetic influences and should not be confused with environmental risk factors, i.e., environmental exposures and lifestyle-related behaviors [30]. This model also provides estimates of cumulative incidence by age as well as familial risk by age, though not for different cancers. The parametric estimates of familial risk were compared to those from the nonparametric model and in turn also adjusted for competing risk of death.

Furthermore, the heritability estimated nonparametrically on the risk scale was obtained in line with the “ACE” sub-model as the ratio of twice the difference in MZ and DZ concordance risk to the total variance in risk based on nonparametric estimates [26,31]. This is not directly comparable to the heritability modelled parametrically on the liability scale [26]. The heritability in risk was estimated by age for overall hematologic malignancies and at age 100 for overall hematologic malignancy vs. other cancer sites.

When illustrating risks and genetic and environmental effects by age, we focused on age 40 and older but did not restrict occurrences by age.

Model sensitivity was assessed by comparing the parametric and nonparametric estimates of cumulative incidence functions and familial risks by age. If there were no substantial differences, we relied on the parametric estimates as they allow for more accurate inference.

All analyses were performed using the statistical software R version 3.5.2 [32] with mets package version 1.2.6 [26,28]. All tests were two-sided and performed at the 5% significance level.

## 3. Results

The study base included 316,397 individual twins from the NorTwinCan cohort; 88,618 (28%) of these were monozygotic (Table 1). A total of 58,449 cancer diagnoses were reported; out of these, 3459 (6%) were hematologic malignancies. The most common types, NHL and leukemia, constituted 44% and 35% of the hematologic malignancies, respectively. MM and HL accounted for 15% and 7%, respectively.

The cumulative incidences by age of hematologic malignancies are shown in Figure 1. There was no substantial variation among cumulative incidences by age of leukemia, MM, and HL across the Nordic countries. However, the cumulative incidences of NHL diverged between the Nordic countries, mainly at high ages (Appendix A). Stratification by sex revealed that males had slightly higher risks for all four cancer types (Appendix A). No difference between cumulative incidences by zygosity was seen (Appendix A).

The lifetime risk (in %, 95% CI) of overall hematologic malignancy, given by cumulative incidence at age 100, was 2.5 (2.4–2.6) (Table 2). For individuals diagnosed with more than one hematologic malignancy during follow-up, time to first diagnosis was used. The lifetime risks for each cancer site were NHL: 1.1 (1.0–1.2), leukemia: 0.9 (0.8–1.0), MM: 0.4 (0.3–0.4), and HL: 0.1 (0.1–0.1). The number of concordant and discordant pairs by zygosity is also provided in Table 2. Discordance, in this case, refers to pairs in which one twin was diagnosed with a hematologic malignancy, and (i) the co-twin was diagnosed with a different cancer or (ii) none at all. Only complete pairs, i.e., pairs where both twins were in the cohort, were included in this table and in the following concordance-related estimates.

Familial risks of hematologic malignancies conditional on the same cancer in the co-twin before age 100 are shown in Table 2. They were elevated for DZ pairs in comparison to the corresponding individual lifetime risks, but the difference was only significant for overall hematologic malignancy, leukemia, and NHL. Furthermore, all familial risks were suggestively higher among MZ than DZ pairs; however, the differences were not significant. There were not enough DZ pairs concordant for HL to provide an estimate of familial risk.

Estimates of tetrachoric correlation (from the biprobit model), and genetic and environmental contributions to variation in liability to hematologic malignancies (from the “ACE” sub-model) are shown in Table 3. The tetrachoric correlations were higher for MZ than DZ pairs, but none of the differences were statistically significant. There were not enough DZ pairs concordant for HL to estimate the tetrachoric correlation.

The genetic and environmental contributions to variation in liability to overall hematologic malignancy and to NHL, MM, and leukemia individually were very similar. Generally, the heritability was moderate, while the contribution from shared environment was practically zero and the remaining variation was attributed to unique environmental effects (Table 3). For overall hematologic malignancy, the estimates and 95% CI were heritability: 0.24 (0.14–0.33), shared environment: 0 (NA–NA), and unique environment: 0.76 (0.67–0.86). The respective values for the specific hematological malignancies are also listed in Table 3. For HL, the heritability was higher, 0.56 (0.32–0.80); the unique environmental contribution equivalently lower, 0.44 (0.20–0.68); and there was no shared environmental contribution.

Relative recurrence (RR) risks and 95% confidence intervals of cross-cancers at age 100 are shown in Figure 2. For example, the risk for leukemia and NHL to cooccur in DZ twin pairs was 2.3 times higher than the risk of these two cancers occurring independently. The RR for DZ twins were generally higher than one indicating a familial predisposition, though only leukemia/NHL and leukemia/leukemia were statistically significant. Furthermore, all of the ratios were higher among MZ than DZ twins, suggesting the influence of a shared genetic component, though none of these differences were statistically significant.

Cumulative incidence and familial risk by age for overall hematological malignancy are depicted in Figure 3. The DZ familial risk lying above the cumulative incidence, especially at old age, indicates a familial predisposition. The MZ familial risk was somewhat higher than DZ, suggesting a genetic component, but this difference was not statistically significant. The curves were estimated using the “AE” submodel as it allowed for more accurate inference than the nonparametric counting process modelling (Appendix A). Curves based on the biprobit model and “ACE” sub-model were almost identical to Figure 3 but with wider confidence intervals.

The genetic (heritability) and unique environmental 
contributions to variation in liability to overall hematologic malignancy by 
age are shown in Figure 4. The 
heritability decreases from around 55% at age 40 and stabilizes around 20–25% 
at age 55. The remaining variation is accounted for by the unique environmental 
contribution as shared environmental contribution (C) was not different from 
zero at any age. 

The heritability in risk of overall hematologic malignancy by age estimated using the nonparametric approach was fairly stable over age and not significantly different from zero (Appendix A). The shared environmental contribution was zero at all ages.

Familial risks for each hematologic malignancy given any hematologic malignancy in the co-twin were estimated for MZ and DZ (in %, 95% CI): NHL: 3.4 (1.1–5.6) and 1.9 (1.0–2.8); HL: 0.7 (−0.2–1.6) and 0.1 (−0.1–0.2); MM: 1.4 (0.3–2.6) and 0.6 (0–1.2); and leukemia: 2.4 (0.7–4.1) and 2.3 (1.2–3.4).

Finally, to analyze the concordance of overall hematologic malignancy and non-hematologic cancers, we counted the number of occurrences by zygosity and estimated lifetime cross-cancer relative recurrence risk and heritability on the risk scale (Appendix A). The relative recurrence risk of overall hematologic malignancy and the following cancers were borderline significantly higher than one among DZ twins, indicating a familial predisposition for colon, lung, prostate, kidney, bladder, skin (melanoma and non-melanoma), brain/central nervous system, and lip/oral cavity/pharynx cancers. However, none of the ratios were significantly higher among MZ twins than DZ and none of the heritabilities were significantly different from zero, providing no evidence of shared genetic influence.

## 4. Discussion

The results from this prospective cohort study of Nordic twins with the longest follow-up ever provide evidence of increased familial risk of hematologic malignancies. For a DZ twin with an affected co-twin, the overall risk (4.5%) of developing a hematologic malignancy was almost twice the risk in the general population (2.5%). Notably, this DZ familial risk was significantly higher than that of the general population already from 40 years of age. The lifetime risks of leukemia and NHL were about twice as high if a DZ co-twin had been diagnosed with any hematologic malignancy. The estimates of relative recurrence risk indicated familial predisposition, not only within the hematologic malignancies but also between overall hematologic malignancy and other types of cancer (as defined by the NORDCAN classification). Moreover, about a quarter (24%) of the variation in liability to develop a hematological malignancy was attributed to genetic effects (heritability) and the remaining variation was explained by effects not shared by twin siblings. The heritability varied across age and was around 55% at age 40 but decreased thereafter to 20–25% and stayed constant at the older ages.

The lifetime risks for individuals were not entirely consistent with the estimates reported by NORDCAN. They provide the following risks of developing cancer in the absence of competing risks of death before age 75 in the Nordic countries for male/female (%); NHL: 1.3/0.9, HL: 0.2/0.2, MM: 0.5/0.3, and leukemia: 0.9/0.6 [1]. The lifetime risks for the corresponding age and sex estimated in this study (Appendix A) were all lower than those from NORDCAN. The differences between the estimates from NORDCAN and our findings might reflect the fact that we adjusted for competing risk of death; omitting this will likely bias estimates upwards. There is a variable age-at-entry of the twins into the cohorts, which may be associated with absence of hematological cancers diagnosed early in life in some cohorts. However, this is taken into account by adjusting for delayed entry, providing valid estimates at all ages.

Several studies have reported evidence of familial predisposition to hematologic malignancies [9,12,13,14,15,16,17,18,19,20,21,22]. One is a study by Fallah et al. [7], who found significantly increased familial risks of NHL for various familial relationships by sex relative to the general population in the Nordic countries. Not surprisingly, the highest familial risk of NHL was found among twins; however, the zygosity was unfortunately not specified. Similar results were seen in a study of familial risk of HL by Kharazmi et al. [9], who also ignored zygosity. The distinction between MZ and DZ twins is considered crucial to the analysis presented in this paper as it allows for the partitioning of genetic and environmental influences.

Another approach was taken by Sud et al. [19], who published results from an extensive study including 153,115 Swedish patients with hematologic malignancies. They utilized the large sample size by examining the interrelationships between familial relative risks (corresponding to the relative recurrence risks of this study) of 22 different hematologic malignancies. To the extent that they are comparable, the results are similar to the ones we provide. A weakness of our study is the lack of power in the sense that we were limited by the low number of concordant twin pairs. Therefore, we consider only the four main hematologic malignancies when assessing familial dependence measures and when estimating familial predisposition by age hematologic malignancy as an overall group. However, the fact that we were able not only to assess the course of familial risk by age but also to partition the variation in liability to hematologic malignancy by age into genetic and environmental factors is a novel contribution.

A common problem regarding studies of familial risk of hematologic malignancies, especially the subtypes, is the questionable validity of self-reported family history [20,33]. As this is a multi-country study based on four nationwide registries with very long follow-up times, we avoided issues of misclassification and recall bias.

The estimated genetic and environmental influences for NHL, HL, and MM based on the NorTwinCan cohort are reported for the first time in this paper. The heritability estimates for NHL, MM, and leukemia were generally low (16–25%), while the shared environmental effects were estimated to be zero. The remaining variance was attributed to exposures and experiences not shared by twin siblings, which includes a wide range of possible factors from occupational exposures and lifestyle factors to somatic mutations and random effects at the cellular level. The estimates of genetic and environmental influences on liability to leukemia were similar to those from Lichtenstein et al. [23] despite using methods that take into account the timing of events and censoring [26].

In contrast to other hematological malignancies, HL showed considerably more heritable variation, with a heritability estimate (56%) that was more than twice as high as the other hematologic malignancies. This finding is congruent with the study by Mack et al. [22]. Excluding HL from overall hematologic malignancies in our analyses did not result in noteworthy differences in familial risk by age or in genetic and environmental effects by age, though this might not have been the case if we focused on childhood and young adulthood.

Though it could be argued that the hematologic malignancies are fundamentally different and should be studied separately, we considered it relevant, from an epidemiological point of view, to look at overall hematologic malignancies when estimating genetic and environmental effects. By doing so we have, for the first time, gained insight into the time-varying influence of genetic and environmental influences on the liability to develop a hematologic malignancy.

We found evidence of familial predisposition to cross-cancer occurrences when comparing overall hematologic malignancy and other cancers, including prostate, lung, and non-melanoma skin cancer. For these cross-cancer associations, the relative recurrence risks were higher among MZ than DZ twins, indicating a possible shared genetic influence, though it was not statistically significant. These are verifications of previous findings; for instance, Goldgar et al. [21] reported evidence of familial predisposition for, e.g., breast cancer/NHL, prostate cancer/NHL, and leukemia/colon cancer and Zhang et al. [12] found evidence of familial predisposition to NHL/lung cancer.

Etiological studies of hematologic malignancies indicate that there are numerous risk factors and that they influence the subtypes to various degrees. For instance, immunodeficiency is deemed a relevant risk factor of NHL along with radiation exposure, alcohol intake, and certain infectious organisms, but we have yet to fully explain the increase of NHL incidence rate that has been seen in developed countries [4,8]. One should keep in mind that many risk factors tend to be aggregated within families, representing shared environmental effects and heritability. Additionally, there are (rare) congenital syndromes, such as Downs syndrome, Fanconi anemia, Diamond Blackfan anemia, and other rare bone marrow failure syndromes, that increase the risk of hematologic malignancy [34]. While an underlying syndromic predisposition could also explain familial clustering of these cancers, it seems unlikely that these known rare syndromes explain our findings. Though the influence of many environmental risk factors has already been studied, it seems there are more to be found. A more recent trend that could influence cancer risk, especially cancers of the lymphatic system, is tattooing. Pathologists have reported cases of pigments from tattoo ink found in lymph nodes [35,36,37], and it has been hypothesized that ink particles may also reach the blood stream and become distributed to organs. In a mouse study, the authors reported that tattoo pigments were detected in the liver but not in other internal organs [38]. However, the (cancer) risk of long-term exposure to tattoo ink has yet to be examined. The matched co-twin design provides the perfect setting for such studies.

## 5. Conclusions

In this largest ever studied twin cohort with the longest follow-up, we found evidence of increased familial risk of hematologic malignancies and indications of familial predisposition, not only within the hematologic malignancies but also between overall hematologic malignancy and other types of cancer. Heritable factors were most important among young and middle-aged adults and decreased across age after 55 years of age. The discovery of decreasing familial predisposition with increasing age underscores the importance of cancer surveillance in families with hematological malignancies.

## Figures and Tables

**Figure 1 cancers-13-03023-f001:**
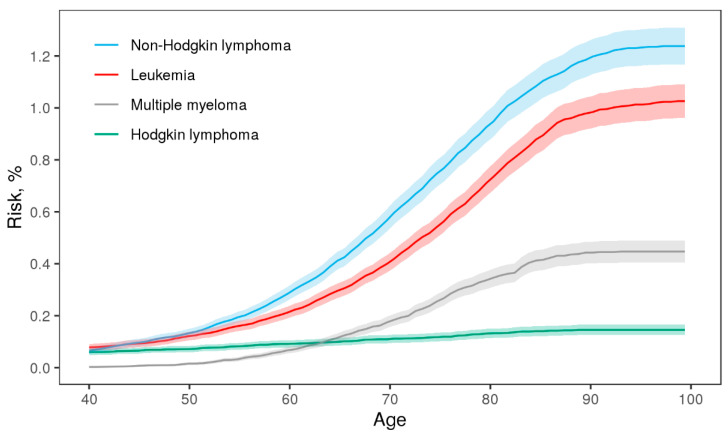
Cumulative incidence and 95% confidence intervals for hematologic malignancies by age in the NorTwinCan cohort, adjusted for censoring and competing risk of death. Incidence before age 40 was very low.

**Figure 2 cancers-13-03023-f002:**
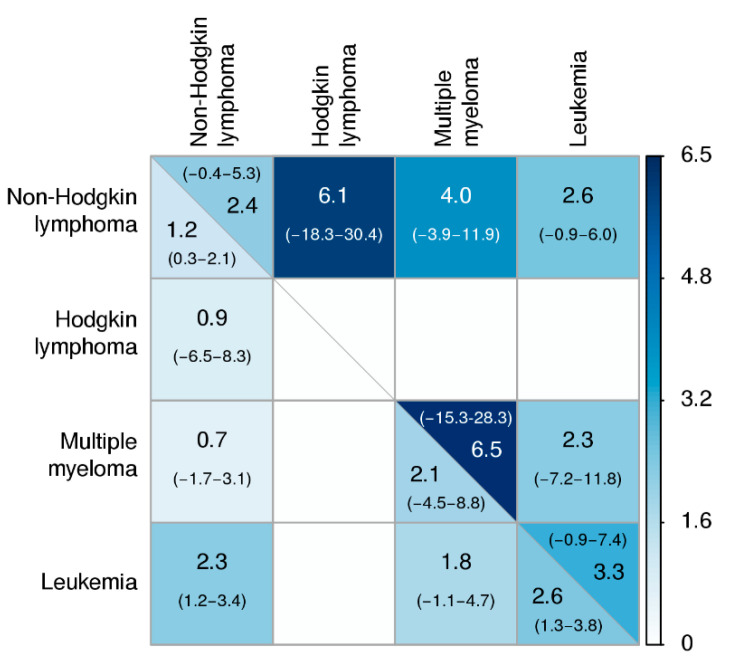
Lifetime cross-cancer relative recurrence risk (concordance risk in pairs relative to independent occurrence risk) and 95% confidence intervals for hematologic malignancies in the NorTwinCan cohort, adjusted for censoring and competing risk of death. Monozygotic pairs are in upper triangle, and dizygotic pairs are in lower triangle. The color scale indicates the magnitude of estimates.

**Figure 3 cancers-13-03023-f003:**
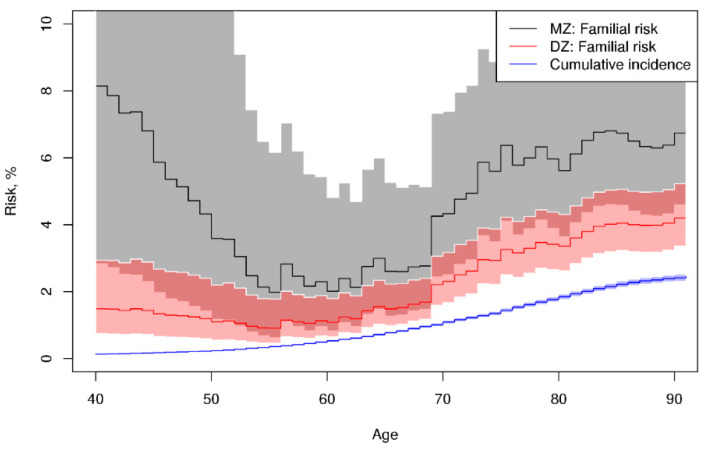
Cumulative incidence and familial risk for monozygotic (MZ) and dizygotic (DZ) twins by age and 95% confidence intervals for overall hematologic malignancy in the NorTwinCan cohort, adjusted for censoring.

**Figure 4 cancers-13-03023-f004:**
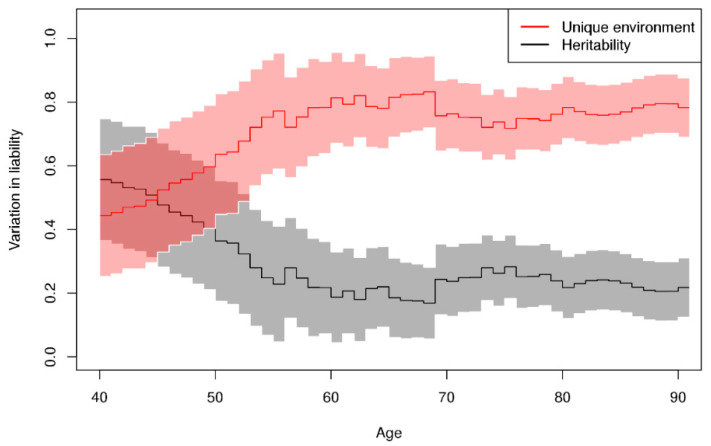
Shared genetic (heritability) and unique environmental contribution to variation in liability to hematologic malignancy by age and 95% confidence intervals in the NorTwinCan cohort, adjusted for censoring.

**Table 1 cancers-13-03023-t001:** Characteristics of the NorTwinCan cohort of 316,397 twins and a total of 3459 incident hematologic malignancies.

Country	Denmark	Finland	Norway	Sweden	Total
Birth Cohort	1870–2004	1875–1957	1915–1991	1886–2008	
*n* individual twins	112,560	31,447	32,332	140,058	316,397
*n* (%) MZ twins	25,120 (22)	8368 (27)	13,100 (41)	42,030 (30)	88,618 (28)
*n* (%) female twins	55,652 (49)	15,717 (50)	17,473 (54)	73,640 (53)	162,482 (51)
First date follow-up	January 1943	February 1974	April 1956	April 1961	
End of follow-up	December 2016	December 2016	December 2015	December 2015	
Median follow-up time (IQR), years	46.6 (26.5–62.1)	41.3 (31.7–41.3)	52.9 (39.4–59.9)	42.0 (22.2–58.2)	41.8 (26.4–58.2)
Median entry age (IQR), years	0 (0–0)	31.8 (25.6–43.1)	2.4 (0–16.3)	1.7 (0–23.3)	0 (0–24.6)
**Number of incident cancers**					
Any cancer site	19,581	7044	5465	26,359	58,449
Non-Hodgkin lymphoma	424	261	184	645	1514
Hodgkin lymphoma	80	29	22	98	229
Multiple myeloma	144	72	53	253	522
Leukemia	406	141	102	545	1194

**Table 2 cancers-13-03023-t002:** Estimates of lifetime and familial risk, and the number of concordant and discordant pairs for hematologic malignancies among twins in the NorTwinCan cohort, adjusted for censoring.

Cancer Site	Lifetime Risk, % (95% CI) ^1^	Number of Twin Pairs						Familial Risk, % (95% CI) ^1^	
		MZ				DZ			
		Conc.	Disc.		Conc.	Disc.		MZ	DZ
			Other ^3^	None ^4^		Other ^3^	None ^4^		
Overall hematologic ^2^	2.5 (2.4–2.6)	21	232	591	30	592	1653	7.6 (4.8–11.8)	4.5 (3.1–6.5)
Non-Hodgkin lymphoma	1.1 (1.0–1.2)	4	115	246	8	279	732	5.2 (1.8–13.9)	2.1 (1.0–4.3)
Hodgkin lymphoma	0.1 (0.1–0.1)	1–3	14	52	0	33	110	7.4 (1.8–25.6)	-
Multiple myeloma	0.4 (0.3–0.4)	1–3	49	94	1–3	92	244	2.6 (0.6–10.0)	1.4 (0.2–9.2)
Leukemia	0.9 (0.8–1.0)	6	72	207	8	219	582	5.4 (2.1–13.3)	3.6 (1.7–7.3)

^1^ Estimated at age 100. ^2^ For individuals who were diagnosed with more than one hematologic malignancy during follow-up (*n* = 34), time to first diagnosis was used. ^3^ Discordant for cancer type, i.e., one twin was diagnosed with the specified cancer and the co-twin was diagnosed with another type of cancer. ^4^ Discordant for cancer, i.e., one twin was diagnosed with the specified cancer and the co-twin did not have cancer.

**Table 3 cancers-13-03023-t003:** Estimates of tetrachoric correlation (biprobit model), and genetic and environmental contribution (ACE model) to variation in liability to hematologic malignancies among twins in the NorTwinCan cohort, adjusted for censoring.

Cancer Site	Tetrachoric Correlations, (95% CI)		Heritability, (95% CI)	Shared Environment, (95% CI)	Unique Environment, (95% CI)
	MZ	DZ			
Overall hematologic ^1^	0.24 (0.12–0.35)	0.12 (0.03–0.20)	0.24 (0.14–0.33)	0.00 (NA–NA)	0.76 (0.67–0.86)
Non-Hodgkin lymphoma	0.28 (0.05–0.48)	0.10 (−0.03–0.22)	0.25 (0.08–0.42)	0.00 (NA–NA)	0.75 (0.58–0.92)
Hodgkin lymphoma	0.57 (0.29–0.76)	-	0.56 (0.32–0.80)	0.00 (NA–NA)	0.44 (0.20–0.68)
Multiple myeloma	0.26 (0.02–0.48)	0.17 (−0.12–0.43)	0.19 (−0.54–0.93)	0.07 (−0.54–0.69)	0.74 (0.50–0.97)
Leukemia	0.31 (0.10–0.49)	0.23 (0.09–0.36)	0.16 (−0.31–0.64)	0.14 (−0.19–0.48)	0.69 (0.50–0.89)

^1^ For individuals who were diagnosed with more than one hematologic cancer during follow-up (*n* = 34), time to first diagnosis was used.

## Data Availability

Restrictions apply to the availability of these data. The data were obtained from the participating twin cohorts and national cancer registries. Requests to access additional data need to be made through the individual national cohorts and registers responsible for the data sets.

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
