# Peer review of "Familial Risk and Heritability of Hematologic Malignancies in the Nordic Twin Study of Cancer"

_cancers, 2021, doi:10.3390/cancers13123023_

Round 1

Reviewer 1 Report

Findings of decreased familial predisposition in older age group is interesting, but I am not agreeing with author's statement in lines 27-28 or 42-43. They may restate, for example,  "surveillance of hematological cancer should not be the same in different age group".  It is not clear what environmental risk was taken into account during this study, as method did not clearly show the data corrected. Authors mentioned in line 146, but I did not understand what  " everything" means.  

Author Response

Dear reviewer, thank you for taking the time to review our manuscript.

Findings of decreased familial predisposition in older age group is interesting, but I am not agreeing with author's statement in lines 27-28 or 42-43. They may restate, for example,” surveillance of hematological cancer should not be the same in different age group".  

This is an excellent suggestion. The lines in question have been changed accordingly.

It is not clear what environmental risk was taken into account during this study, as method did not clearly show the data corrected. Authors mentioned in line 146, but I did not understand what  " everything" means.  

Thank you for pointing this out. We have rephrased and elaborated on the description of environmental effects and hope it has become clearer.

Reviewer 2 Report

This is an attractive study that addresses an area of significant interest. Much of the data either confirms previously published results and/or does not reach statistical significance, which reduces the impact somewhat. In addition, the methods are weakened by the inability to further sub-classify malignancy diagnoses. However, the findings remain important and do add to our knowledge of familial risk of hematologic malignancy.

Additional comments:

  1. Figure 2 as presented is rather confusing and difficult to interpret. Depiction of overall hematologic malignancy risk, and if possible risk of other cancers, would be more informative.
  2. Excluding cancer incidence younger than age 40 is a major flaw since familial hematolgic malignancy very commonly presents earlier than this age. 
  3. The purpose of Figure 1 is unclear since there is no comparison to any control cohort. 
